# Identification of Genomic Variants and Candidate Genes for Reproductive Traits and Growth Traits in Pishan Red Sheep Using Whole-Genome Resequencing

**DOI:** 10.3390/biology14060636

**Published:** 2025-05-30

**Authors:** Maimaitijiang Muhetapa, Mengting Zhu, Aladaer Qi, Sulaiman Yiming

**Affiliations:** College of Animal Science, Xinjiang Agricultural University, Urumqi 830000, China; muhtabarmamatjan@163.com (M.M.); zhumengting@xjau.edu.cn (M.Z.); aqi@xjau.edu.cn (A.Q.)

**Keywords:** Pishan red sheep, reproductive trait, selection signature, growth trait, GWAS

## Abstract

Sheep breeding plays a key role in agriculture, but efficiently improving traits like reproduction and growth remains a challenge. Pishan red sheep, a unique breed known for year-round fertility, high birth rates, and strong genetic traits, offer valuable insights into these traits. This study explored the genetic basis of their superior qualities by analyzing their entire DNA. Researchers identified over 53,968,686 high-quality SNPs and discovered 90 key genes linked to growth, reproduction, and disease resistance. Notably, genes such as BMPRIB and HPGDS were found to influence litter size through specific biological pathways. Additionally, 59 genetic markers tied to growth (e.g., body size) and reproduction (e.g., fertility) were pinpointed. These findings help to explain why Pishan red sheep excel in these traits and provide practical tools for farmers and breeders to select animals with desirable characteristics more accurately. By applying this knowledge, breeding programs can develop healthier, more productive sheep, boosting agricultural efficiency and supporting sustainable farming practices.

## 1. Introduction

Reproductive traits are critically important economically in sheep production, and enhancing production efficiency serves as a key driver for advancing the sustainable development of the sheep breeding industry [1,2]. Growth traits are key indicators of sheep growth, development, and individual breeding potential. Identifying genes influencing sheep reproduction and growth is critical for the rapid development of the sheep industry and is a key focus of current research [3]. Pishan Red Sheep, an indigenous breed characterized by distinctive brown-red or light-black coats, are predominantly found in Hotan’s Pishan, Moyu, Yutian, Qira, and Lop Counties of Xinjiang. Core breeding zones cluster around Pishan County’s Muji Town and Qiaoda Township. At the end of the 2023, the registered population stood at 357,600 head. Under free-range farming practices, mature rams attain an average live weight of 76.3 kg, compared to 52.2 kg for adult ewes. Sexual maturity is achieved at 12 months for males and 10 months for females, with reproductive cycles featuring 20-day estrous intervals and 145–156-day gestation periods. The majority of ewes demonstrate triennial parturition capacity (three lambing cycles per two years), while intensive management systems elevate litter productivity to 181.44%. Notably, they exhibit year-round reproductive cyclicity, sustained high fecundity across breeding cycles, and remarkable genetic stability under environmental stressors. Their robust adaptability is further demonstrated through drought tolerance, disease resistance, and efficient nutrient utilization in low-input grazing systems. Crucially, these sheep maintain consistent reproductive performance even under suboptimal ecological conditions, a trait attributed to their evolutionary adaptation to arid plateau environments. This unique combination of prolificacy, environmental hardiness, and genetic conservatism positions Pishan red sheep as a valuable natural model for investigating the genomic and epigenetic mechanisms underlying hyperprolificacy in meat sheep breeds.

With the development of sequencing and molecular marker technologies, the use of genome-wide association studies (GWASs) has become an effective strategy for mining genes related to important economic traits in the field of sheep population genomics [4]. Lan et al. reported on the growth and development basics of double lambs of Pishan red sheep and found that the tails of Pishan red sheep grew fastest between birth and 1 month of age. Additionally, fitting and regression analysis demonstrated that the von Bertalanffy model was the best-fitting model [5]. Xu et al. [6] performed a GWAS on the tail types of large-tailed and small-tailed Han sheep, and found an enrichment of significant single-nucleotide polymorphisms (SNPs) on both autosomes and sex chromosomes. Enrichment analysis showed that the *CREB1*, *STEAP4*, *CTBP1*, and *RIP140* genes were related to animal fat metabolism and cell development. Additionally, based on whole-genome resequencing technology, the population genetic structure and the selection signal region of the litter size trait of Pishan red sheep were investigated, and the relevant candidate genes were verified. Meanwhile, Lv et al. performed whole-genome resequencing for domestic sheep and their wild relatives, and used the resulting data to reconstruct the phylogenetic and evolutionary history of sheep species, as well as to explore the characteristics of gene exchange and domestication selection among the species. The results revealed that European mouflons might have arisen through hybridization events between a now-extinct sheep in Europe and feral domesticated sheep 5000 to 6000 years ago. The authors also detected introgression events that resulted in the introduction of genes related to nervous response (*NEURL1*), neurogenesis (*PRUNE2*), hearing ability (*USH2A*), and placental viability (*PAG11* and *PAG3*) from other wild species into domestic sheep and their ancestral wild species [7]. Genomic regions harboring genes associated with distinct morphological and agronomic traits that might be past and potential future targets of domestication, breeding, and selection were also identified. Moreover, the authors identified *PDGFD* as a likely causal gene for fat deposition in the tails of sheep through omics [8]. Combined, these findings provide insights into the demographic history of sheep and provide a valuable genomic resource for future genetic studies, as well as contributing to improving genome-assisted breeding of sheep and other domestic animals [8]. Similarly, through a comparative analysis of the whole genomes of modern domestic sheep and wild populations, Dou et al. [9] determined that the strongest selected signal region in the whole genome of goats was located in an 80 kb region on chromosome 15, and confirmed that two protein-coding genes, namely *STIM1* and *RRM1*, were associated with neurotransmitter transport and neural tube development during the embryonic period, and that the selected region was related to domestication behavior [9]. Additionally, which genes are involved in the regulation of sheep reproductive traits and the relevant underlying mechanisms remain incompletely understood.

Therefore, to clarify the genetic basis of the reproductive and growth traits of Pishan red sheep, we undertook whole-genome resequencing of 219 Pishan red sheep native to Xinjiang and subsequently analyzed the genetic variation and the genetic structure of the population, and we sought to identify the selection regions and candidate genes for growth and reproductive traits. Furthermore, through a GWAS, we identified genes and SNPs important for the reproductive and growth traits of these sheep, thereby providing a basis for the creation of a new germplasm of native sheep.

## 2. Materials and Methods

### 2.1. Ethics Approval

All animal-related experimental procedures were performed according to the Regulations for the Administration of Affairs Concerning Experimental Animals of China, and were approved by the Animal Care Committee of Xinjiang Agricultura University (approval number: 2024004), which is responsible for overseeing the ethical use of animals in research within the university. All methods are reported in accordance with ARRIVE guidelines for the reporting of animal experiments.

### 2.2. Sample Collection and Whole-Genome Resequencing

A total of 219 healthy native Xinjiang Pishan red sheep (Figure 1), aged 3–4 years and of a similar weight, were selected from Xinjiang Xiyu Muyangren Agriculture and Animal Husbandry Technology Co., Ltd. According to records of their litter size, 96 sheep were allocated into a high-fecundity (HF, ewes who had given birth to two or more sets of twins in a row, 54.78 ± 8.12 kg) group and 123 sheep into a low-fecundity (LF, ewes who had given birth to a single lamb at least twice in a row, 53.96 ± 7.17 kg) group. Phenotypic data on growth traits (i.e., body weight, body height, body length, chest circumference, cannon circumference, tail length, tail width, and tail thickness) were collected in accordance with the provisions of NY/T 1236. DNA was isolated from the blood of the sheep using the TIANamp Blood DNA Kit (Tiangen Biochemical Technology (Beijing) Co., Ltd., Beijing, China), according to the manufacturer’s instructions, and its quality was evaluated by 1% agarose gel electrophoresis. DNA purity and concentration were assessed with a NanoDrop 2000 spectrophotometer (Thermo Fisher, San Diego, CA, USA). The DNA was fragmented by ultrasonication (Bioruptor Pico sonicator), resulting in fragments approximately 350 bp in length. Libraries were constructed according to the manufacturer’s instructions (Illumina, San Diego, CA, USA) and then paired-end-sequenced at 10× coverage on the Illumina HiSeq2500 platform (Illumina, San Diego, CA, USA) by BGI Tech Co., Ltd. (Shenzhen, China).

### 2.3. Quality Control and Read Mapping

To ensure the reliability of subsequent analyses, the raw data generated by high-throughput sequencing were quality-filtered using fastp software (v.0.19.4) [10], yielding clean reads that could be used for analysis with default parameters. Then, the BWA-MEM algorithm (v.0.7.17) was used to map the clean sequencing reads to the sheep reference genome (Oar_v4.0), followed by identification of polymorphic sites using SAMtools software (v.1.2) [11]. SAMtools and PICARD (v.2.18.17) were used to remove duplicates. Variant detection was performed through a dual-calling pipeline to ensure accuracy. Initial SNP and InDel identification was conducted using GATK’s HaplotypeCaller (v3.8.1) with default parameters. Following alignment refinement, parallel variant calling was executed through SAMtools (parameters: -q 1 [minimum mapping quality] -C 50 [adjust mapping quality] -m 2 [minimum number of alternate alleles] -F 0.002 [indel fraction threshold] -d 1000 [maximum read depth]). The combined variant calls underwent stringent quality filtering using VCFtools (v0.1.13), with thresholds set to QUAL > 30, DP > 10, and MQ > 40. SNPs with a Minor Allele Frequency (MAF) of <0.05 and a Hardy–Weinberg equilibrium *p*-value of <1 × 10^−5^ (--maf 0.05 --hwe 1 × 10^−5^) were filtered out. The filtered SNP data were annotated using ANNOVAR v.2013-06-21. The Haplotyper and GVCFtyper data were output as VCF files, and the obtained data were filtered using PLINK software (v.1.9).

### 2.4. Population Genetic Diversity Analysis

Before conducting the analysis, all SNPs were pruned using the indep-pairwise function of PLINK v.1.09 software [12,13]. Principal Component Analysis (PCA) was conducted using EIGENSOFT [14]. ADMIXTURE (v.1.3) software was employed to construct the population genetic structure, with *K*-values ranging from 2 to 4. Linkage disequilibrium analysis was carried out with PopLDdecay (https://github.com/BGI-shenzhen/PopLDdecay (accessed on 26 January 2024)).

### 2.5. Analysis of Genome-Wide Selective Sweep Regions

The fixation index (*F*_ST_) and nucleotide diversity (π) were calculated with VCFtools v.0.1.14; selective sweep analysis was conducted using a sliding-window method (150 kb windows, steps of 75 kb). Overlapping candidate selective regions with *F*_ST_ and pi ratio values in the top 5% were considered to be under strong selective sweep, and were examined for potential candidate genes. Finally, the candidate genes were subjected to Gene Ontology (GO) and Kyoto Encyclopedia of Genes and Genomes (KEGG) enrichment analysis, which was performed in R software (v.4.3.3) [15,16].

### 2.6. GWAS

An analysis of the association between growth traits (body weight [BW], body height [BH], body length [BL], chest circumference [CC], cannon circumference [TC], tail length [TL], tail width [TW], tail thickness [TT]) and reproductive traits (litter size [LS], birth weight [LW]) was performed using GLM, MLM, and Farm CPU models in GAPIT3 [17,18]. A GWAS on the growth and reproductive traits of 219 Pishan red sheep was performed using whole-genome resequencing data. SNP sites with MAFs of less than 0.05, those with deletion rates greater than 0.10, and those with markers deviating from HWE at *p* < 1 × 10^−5^ (--maf 0.05 --geno 0.1 --hwe 1 × 10^−5^) were excluded from the analysis. After quality control, 22,551,702 high-quality SNPs were obtained for post-analysis.

To avoid potential false positives in multiple comparisons, the genome-wide significance threshold was adjusted using the Bonferroni test [19]. The threshold for determining a significant association between an SNP and a trait was set to 0.05/effective SNP. Such loci were considered to make important contributions to economic traits in sheep. Furthermore, the quantile–quantile (Q–Q) plots of the GLM, MLM, and Farm CPU for economic traits were implemented in GAPIT. The following models were used in the analysis:(1)GLM model: y=Xa+Zb+e(2)MLM model: y=Xα+Zβ+Wμ+e(3)$$\mathrm{Farm}\;\mathrm{CPU}\;\mathrm{model}\text{: }y_{p}=M_{p1}b_{1}+M_{p2}b_{2}+...+M_{pn}b_{n}+S_{pq}d_{q}+e_{i}$$
where y represents the phenotypic value, y_p_ is the observed value of the pth individual trait, *X* is the fixed-effects matrix, *α* is the fixed-effects vector, *Z* is the SNP typing vector, b and *β* are the SNP marker vectors, *W* is the random-effects matrix, *µ* is the random effect, e is the random residual, M_p1_, …, M_pn_ are the association sites of n genotypes added to the model, b_1_,…, b_n_ are the effect values corresponding to the association sites, S_pq_ is the q genotype corresponding to the pth individual, d_q_ is the effect value of S_pq_, and e_i_ is the residual vector.

Finally, SNP variations were annotated using ANNOVAR v.2013-06-21 based on the sheep reference genome (Oar v.4.0). The identified SNPs were annotated as the closest genes, with a nearest-gene distance of 200 kb. These genes were defined as candidate genes. KOBAS software (http://bioinfo.org/kobas) was used to test the significance of candidate gene enrichment in the KEGG pathway analysis [20,21,22].

## 3. Results

### 3.1. Sequencing, Mapping, and SNP/InDel Variation Annotation

Whole-genome resequencing of the 219 local Pishan red sheep from Xinjiang (Appendix A) generated a total of 9084.81 Gb of raw data, with an average depth of 14.46× (10.45×~20.89×) and an average genome coverage of 97.49%. The GC content was 42.14~45.56% and the mapping rate after alignment with the reference genome reached more than 99%, indicating that the data were of high quality and could be used for further in-depth analysis. Finally, 53,968,686 SNPs and 1,415,272 InDels were obtained after mapping with SAMtools (Appendix A). Statistical analysis of the SNPs showed that variants mainly occurred in intergenic intervals, followed by intronic intervals and exonic intervals. This observation implied a relatively balanced distribution of SNPs across the population, indicative of a normalized genomic population structure. These findings established a solid foundation of reliable data for further investigations into the population structure, as well as the identification of potential selection signals.

### 3.2. Population Genetic Structure and Linkage Disequilibrium

A total of 22,551,702 high-throughput SNPs were obtained after filtering. The SNPs displayed different degrees of distribution on the chromosomes (Figure 2A), and further analysis of population genetic structure. The LF group (low-fecundity) could not be distinguished from the HF group (high-fecundity) based on the PCA (Figure 2B) or population genetic structure analysis (Figure 2C). The HF group had a slower decay rate than the LF group; however, the distance decay was almost the same between the two groups (Figure 2D). Given that the two groups had the same origin, the differentiation between the two was small, which was in line with the actual breeding situation.

### 3.3. Selective Imprints of Reproductive Traits in Pishan Red Sheep

The selected regions for reproductive traits in Pishan red sheep were studied at the genome-wide level. The top 5% *F*_ST_ values and π ratios were selected as the threshold for drawing the selected loci on autosomes between the two populations. The windows appearing in the regions of *F*_ST_ = 0.007287253 (Figure 3A and Appendix A) and π ratio = 1.047011 (Figure 3B) were identified as candidate windows. A total of 98 selected regions were screened, and annotation was performed using ANNOVAR software. The selected window was found to contain a total of 90 positional candidate genes, including 29 in the HF group and 61 in the LF group (Figure 3C,D and Appendix A), with 10 (e.g., *BMPR1B*, *TSHR*, *HPGDS*, *UNC5C*, *PDLIM5*, and *GRID2*) previously reported to be related to reproductive traits. GO enrichment analysis indicated that the candidate genes were mainly associated with neuron differentiation, generation of neurons, neurogenesis, neuron projection development, nervous system development, cell projection organization, and cell development in the Biological Process category (Figure 4A). Additionally, KEGG analysis showed that the candidate genes were mainly enriched in the TGF-beta signaling pathway, the thyroid hormone signaling pathway, pentose and glucuronate interconversions, and signaling pathways regulating the pluripotency of stem cells, and were significantly associated with reproductive traits, as well as with other economic traits (Figure 4B).

### 3.4. GWAS of Growth and Reproductive Traits in Pishan Red Sheep

We first analyzed the phenotypic data (BW, BH, BL, CC, TC, TL, TW, TT, LS, and LW) for the 219 Pishan red sheep, and found that there were marked differences in body characteristics between the LF and HF groups. The standard deviation and coefficient of variation were obtained to describe the degree of dispersion of the data. We observed that the values for BL, TL, and TT were higher in the LF group than in the HF group, while the LS, LW, and BH values displayed the opposite trend (Figure 5).

Subsequently, we carried out a GWAS, GLM model, MLM model, and Farm CPU model using the GAPIT package in R. The results of the three models were consistent, so in this manuscript, the MLM result is shown in the results. After adjusting the significance threshold for the GWAS, significant SNPs were detected for three growth traits (BH, CC, and TL) (Figure 6B,D,F). There were no significant SNPs identified for BW (Figure 6A), BC (Figure 6C), TC (Figure 6E), TW (Figure 6G), and TT (Figure 6H). Regarding growth traits, 17 significant loci were found to be associated with BH, and the other four genes were annotated as *PROM1*, *TAPT1*, *LDB2*, and *RUNX2*. Five loci were significantly associated with CC, and these were annotated to seven genes (*KIF16B*, *SNRPB2*, *OTOR*, *FGA*, *FGB*, *PLRG1*, and *DCHS2*) (Figure 6D). In total, 17 SNPs were associated with TL, all of which were located on chromosome 15 and were annotated to the *CADM1* positional candidate genes (Figure 6F). For reproductive traits, 12 significant loci were observed to be associated with LW, which were annotated to 25 positional candidate genes, including *DNAH1*, *NISCH*, *NT5DC2*, and *WDR82* (Figure 6I). There were eight significant loci associated with LS, and they were annotated to 10 positional candidate genes, including *ASPA*, *RAP1GAP2*, *PHIP*, *IRAK1BP14*, *BFSP1*, and *PCSK2* (Figure 6J and Table 1).

### 3.5. Candidate Gene Enrichment Analysis

GO enrichment analysis was performed on the selected candidate genes, with the results showing that these genes were mainly enriched in cell adhesion molecule binding, macromolecular complex binding, tissue homeostasis, negative regulation of the apoptotic signaling pathway, cellular protein complex assembly, cell growth, growth, and regulation of neuron differentiation, among other processes. Meanwhile, KEGG pathway enrichment analysis of the candidate genes for growth traits indicated that they were mainly enriched in histidine metabolism and platelet activation. For reproductive traits, the candidate genes were mainly associated with base excision repair, the calcium signaling pathway, and histidine metabolism (Figure 7).

## 4. Discussion

As a pivotal hub along the Silk Road Economic Belt and a key livestock production base in Northwest China, Xinjiang has developed distinctive sheep breeds through long-term selective breeding, characterized by exceptional fecundity, superior meat yield, and remarkable environmental adaptation. Among these, the indigenous Pishan red sheep exemplify evolutionary advantages, including polyestrous cyclicity, multiparity potential (average litter size: 1.8 ± 0.3), and ecological resilience under extreme arid conditions. Nevertheless, the genomic determinants underlying their economically vital traits remain largely unexplored. Our study systematically characterizes the genetic basis of growth and reproductive traits in Pishan red sheep through integrated genomic analyses. The identification of 92 selection signals and 59 trait-associated SNPs provides critical insights into their high fecundity and environmental adaptability. Key positional candidate genes such as BMPRIB and GRID2, previously linked to ovulation rate and neuroendocrine regulation in sheep, were prioritized through selective sweep and GWAS approaches, corroborating their conserved roles in prolificacy. Functional enrichment of candidate genes in the TGF-beta and thyroid hormone signaling pathways aligns with known biological mechanisms coordinating reproductive cyclicity and metabolic adaptation. While the GWAS results revealed distinct SNP sets associated with growth (39 SNPs) and reproductive traits (20 SNPs), the overlap of selection signals between LF and HF groups underscores shared genetic components underlying these economically vital characteristics. These causes may be due to environmental and other non-genetic factors, we will study further in the future. These findings establish a foundational framework for understanding the genomic architecture of Pishan red sheep, offering practical markers for selective breeding programs aimed at enhancing both litter size and production efficiency in arid-region sheep husbandry.

Reproductive traits are of key economic importance in sheep. The economic benefits of lambs in sheep populations with high reproductive performance are 2 to 3 times greater than those in sheep populations with low reproductive performance [40]. The characteristics of perennial estrus and high prolificacy render Pishan red sheep a good animal model for studying the genetic basis of reproductive traits. In this breed, the estrous cycle lasts for approximately 20 days, and the pregnancy cycle lasts for between 145 and 156 days. Most ewes can achieve three births in 2 years, and the lambing rate is 141.44% under large-scale farm feeding conditions. We have previously reported the basis of the growth and development of Pishan red sheep lambs before weaning. We observed that there was a significant difference in daily weight gain between male and female lambs between birth and 30 days of age. We also investigated the growth and development rule of double-lambing, and found that the von Bertalanffy model was the best-fitting model [5,41].

We further observed that tail fat traits had an important effect on body weight. Genome-wide resequencing provides an opportunity to analyze the genetic basis of important economic traits, such as reproduction, during the artificial selection and domestication of animals. Naval-Sanchez et al. used whole-genome resequencing to detect genes under selection during sheep domestication and subsequent artificial selection. To identify selection sweeps, the authors compared genome sequences from 43 modern breeds (*Ovis aries*) and their Asian mouflon ancestor (*O. orientalis*). Then, genes important for agronomic traits in sheep during domestication were screened out, including *SOCS2*, which was found to be related to body weight and milk yield; *ITCHASIP*, which was associated with coat color; and *VEGFA*, which was related to reproduction. Functional enrichment analysis further indicated that the biological changes resulting from domestication and artificial selection were manifested as alterations in body size, fat metabolism, and time to sexual maturity [42]. Similarly, Zhu et al. resequenced the genomes of four representative sheep breeds in northwest China and revealed the molecular mechanism underlying the regulatory roles of the *PAK1*, *CYP19A1*, and *PER1* genes in seasonal reproduction in sheep, thereby providing insights into the microevolution of native sheep and valuable genomic information for identifying genes associated with important reproductive traits in these animals [43]. Lv et al. generated a comprehensive genome resource for wild ovine species, landraces, and improved breeds of domestic sheep, comprising high-coverage (~16.1×) whole-genomes of 810 samples from seven wild species and 158 diverse domestic populations. They explored the genetic basis of wool fineness and identified a novel mutation (chr25: T7,068,586C) in the 3′-UTR of *IRF2BP2* as a plausible causal variant for fleece fiber diameter. The results refined the understanding of genome variation as shaped by continental migrations, introgression, adaptation, and selection of sheep [7]. In this study, we performed a whole-genome resequencing analysis of Pishan red sheep. We found that several positional candidate genes for reproductive traits, including *BMPR1B*, *TSHR*, *HPGDS*, *UNC5C*, *PDLIM5*, and *GRID2*, were mainly enriched in the TGF-beta signaling pathway, the thyroid hormone signaling pathway, and pentose and glucuronate interconversions. These genes have previously been reported to be related to reproductive traits [44,45,46]. TSHR regulates the dynamic balance of thyroid hormones by inducing the expression of the type 2 deiodinase (*DIO2*) and *DIO3* genes in the hypothalamus, thereby regulating the expression of the *GNRH* gene and ultimately affecting the lambing ability of sheep [47]. However, how these genes regulate sheep reproductive traits requires further detailed investigation.

Here, a total of 47 positional candidate genes were found to be associated with growth and reproductive traits through a GWAS based on whole-genome resequencing data. These genes were shown to be related to growth and development, bone development, and embryonic development. Symoens et al. [24] found that defects in the *TAPT1* gene can lead to complex fatal chondrodysplasia, implying that this gene may be related to bone development in Pishan red sheep, and indirectly affect body structure and body size. Dou et al. identified *LDB2* as an important candidate gene for body weight in broilers by GWAS-based analysis [48]. Given that we found that *LDB2* was associated with BH in this study, this suggests that *LDB2* may affect the body weight of Pishan red sheep by regulating body height-related traits. Similarly, the KIF16B/Rab14 molecular motor complex plays a key role in early embryonic development, including organ formation, by regulating the transport of fibroblast growth factor (FGF) receptors [49]. Additionally, Kleinridders et al. demonstrated that the loss of PLRG1 in mice was embryonic-lethal, and that PLRG1 is an important regulator of p53-dependent cell cycle progression. These observations emphasized the important role of PLRG1 in regulating basic cell processes and early embryo development in mice [50]. The *RUNX2* gene can promote the osteogenic differentiation of umbilical cord blood mesenchymal stem cells, thus stimulating bone repair [51]. This suggests that *RUNX2* may be related to skeletal growth and development in Pishan red sheep, and may influence body structure by affecting skeletal development. Interestingly, *BMPR1B* has previously been proposed as a marker gene for litter size in sheep [51,52]. Zhu et al. reported that genes such as *TSHR* may be involved in the regulation of sheep reproductive traits, which is consistent with the results of this study [26].

## 5. Conclusions

In conclusion, in this study, genetic variation detection, LD analysis, and selection signal analysis were performed using whole-genome resequencing. Additionally, candidate genes affecting litter size traits in Pishan red sheep were analyzed. Through a GWAS, an association analysis for growth traits and reproductive traits was undertaken, and several important genes and SNPs affecting these traits in Pishan red sheep were screened out and identified. However, this breed’s genetic regulation mechanism remains to be further studied. Our findings lay a foundation for an in-depth understanding of the germplasm characteristics of Pishan red sheep.

## Figures and Tables

**Figure 1 biology-14-00636-f001:**
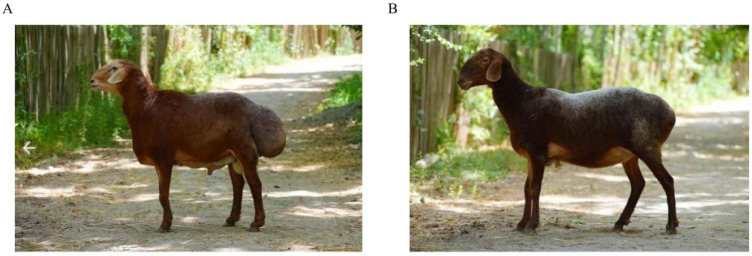
The body appearance characteristics of a ram (**A**) and a ewe (**B**) of the Pishan red sheep breed.

**Figure 2 biology-14-00636-f002:**
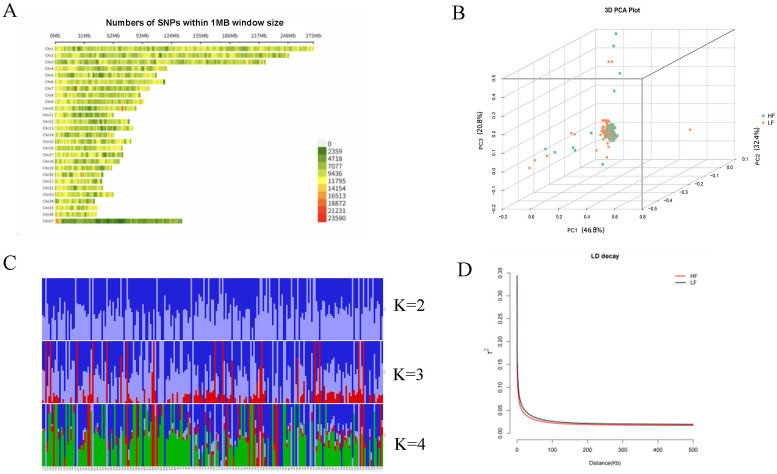
Population genetic structure and LD decay. (**A**) Distribution of SNPs on chromosomes. (**B**) PCA. (**C**) Structure result. (**D**) LD decay. Note: HF means high-fecundity, LF means low-fecundity.

**Figure 3 biology-14-00636-f003:**
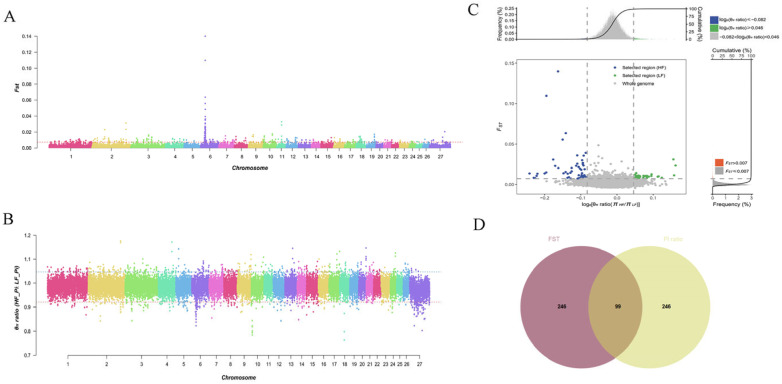
(**A**) Distribution of *F*_ST_ on autosomes of LF and HF of Pishan red sheep. (**B**) Distribution of Pi ratio on autosomes of LF and HF of Pishan red sheep. (**C**) Selection region and selection signal. (**D**) Intersection results of *F*_ST_ and Pi ratio methods. Note: Manhattan threshold line *F*_ST_ = 0.007287253 and π ratio = 1.047011.

**Figure 4 biology-14-00636-f004:**
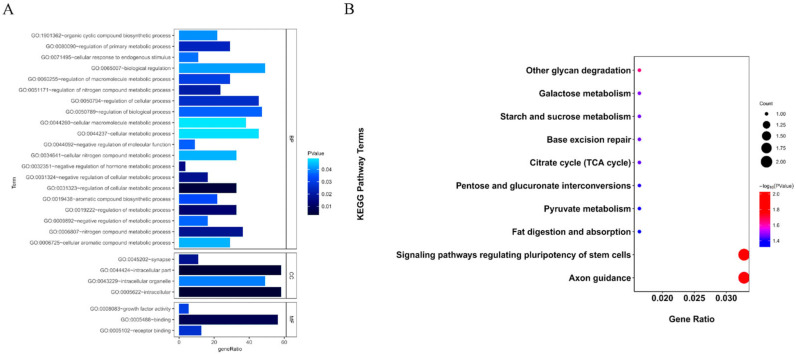
Enrichment results of candidate genes of litter size trait of Pishan red sheep. (**A**) GO Enrichment. (**B**) KEGG pathways.

**Figure 5 biology-14-00636-f005:**
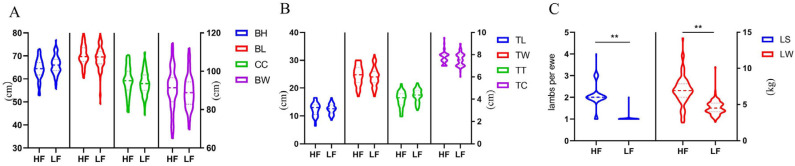
Violin plot of growth traits (**A**,**B**) and reproductive traits (**C**) of Pishan red sheep. Note: body weight (BW), body height (BH), body length (BL), chest circumference (CC), cannon circumference (TC), tail length (TL), tail width (TW), tail thickness (TT), litter size (LS), birth weight (LW). Note: ** represents difference was extremely significant *p* < 0.01.

**Figure 6 biology-14-00636-f006:**
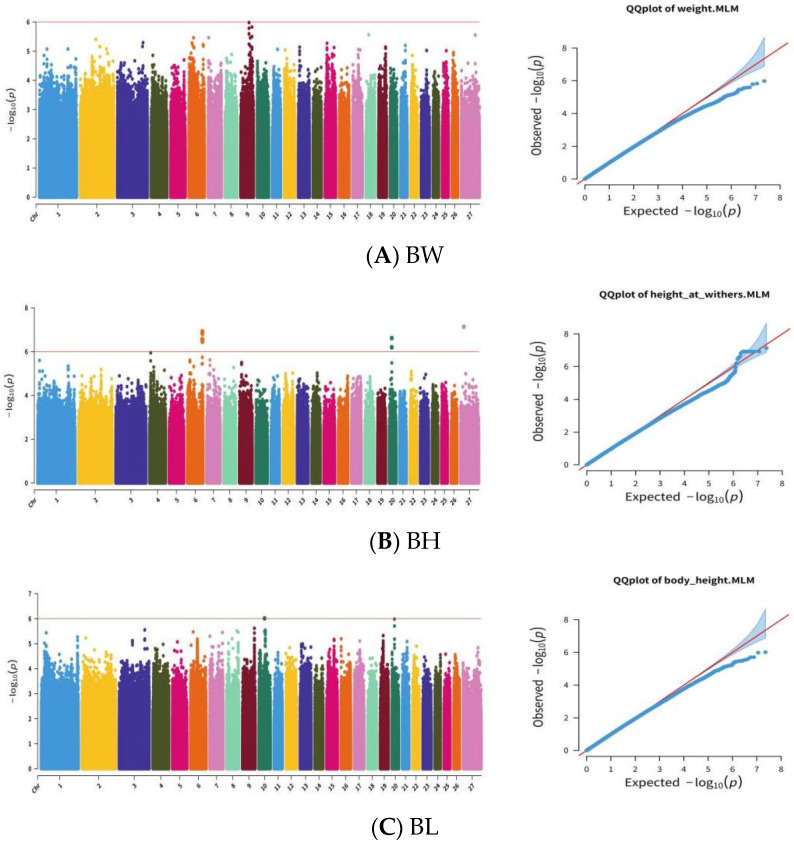
Manhattan and QQplots showing GWAS results for growth traits (BW (**A**), BH (**B**), BL (**C**), CC (**D**), TC (**E**), TL (**F**), TW (**G**), and TT (**H**)) and reproductive traits (LW (**I**) and LS (**J**)).

**Figure 7 biology-14-00636-f007:**
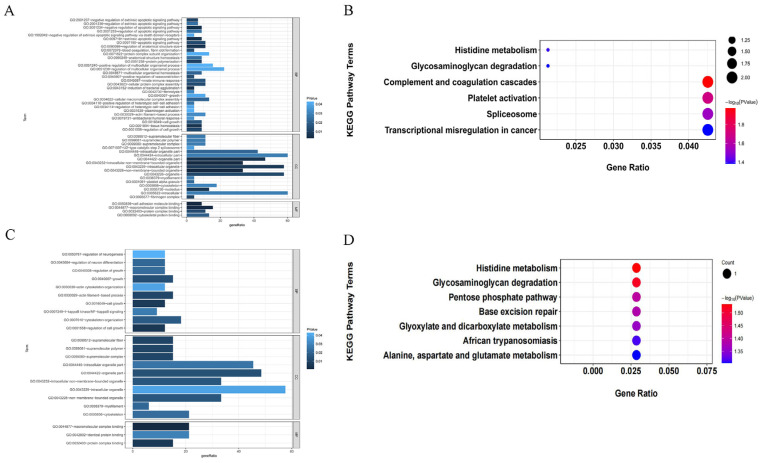
Enrichment results of candidate genes for growth traits and reproduction traits of Pishan red sheep. (**A**) Candidate genes for growth traits from GO enrichment analysis. (**B**) Candidate genes for growth traits from KEGG pathway analysis. (**C**) Candidate genes for reproduction traits from GO enrichment analysis. (**D**) Candidate genes for reproduction traits from KEGG pathway analysis.

**Table 1 biology-14-00636-t001:** GWAS results of growth and reproductive traits in Pishan Red Sheep.

Trait	Chr	POS (Mb)	*p* Value	Genes	Function
BH	6	110,844,020–110,844,500	2.92 × 10^−7^	PROM1, TAPT1	Nipple size Hair follicle development [23]; skeletal development [24]
		110,844,501–110,845,000	1.19 × 10^−7^	TAPT1, LDB2	Skeletal development [24]
		110,845,001–110,847,000	3.46 × 10^−7^	TAPT1, LDB2	Skeletal development [24]
	20	18,934,759–18,934,991	2.39 × 10^−7^	RUNX2	Bone development, osteogenic differentiation [25]
CC	13	10,002,600–10,002,800	6.79 × 10^−7^	KIF16B, SNRPB2, OTOR	Fat deposition [26]
	17	2,870,800–2,871,000	3.36 × 10^−7^	FGA, FGB, PLRG1, DCHS2	Early embryonic development [27]; organ development [28]
TL	15	25,526,000–25,540,000	1.18 × 10^−8^	CADM1	Bone development [29]
LW	2	6,990,000–9,780,000	4.68 × 10^−8^	TRIM32, COL27A1, AMBP, ZNF618	Early growth and development, osteogenic development [30]; lamb birth weight [31]
	3	122,161,200–122,161,300	5.14 × 10^−7^	MGAT4C	Neurodevelopment [32]
	7	24,044,700–24,044,800	5.43 × 10^−7^	PARP2, TTC5	Germ cell development [33]
	14	57,979,500–57,979,600	1.66 × 10^−8^	DPRX, ZNF331	Immunoregulation [34]
	19	48,471,500–48,471,600	5.43 × 10^−7^	DNAH1, NISCH, NT5DC2, WDR82, PHF7, SEMA3G, GLYCTK, STAB1, TLR9, BAP1, TNNC1, PPM1M, TWF2	Embryonic development [35]
	27	49,928,000–49,929,000	4.01 × 10^−7^	GPR173, TSPYL2, KDM5C	Follicular development [36]
LS	8	5,677,600–5,677,700	7.14 × 10^−7^	PHIP, IRAK1BP1	Follicular development [36]
	11	23,580,200–23,584,400	6.73 × 10^−7^	ASPA, RAP1GAP2	Litter size [37]
	13	36,904,100–36,904,200	3.24 × 10^−7^	BFSP1, PCSK2, RRBP1, DSTN	Neuronal development [38]
	27	81,156,000–110,377,700	2.88 × 10^−8^	IDS, DOCK11	Immunodeficiency [39]

## Data Availability

The original contributions presented in the study are included in the article; further inquiries can be directed to the corresponding author/s. The datasets presented in this study can be found in online repositories: the CNGB Sequence Archive (CNSA) of the China National GeneBank DataBase (CNGBdb), under accession number CNP0005646s.

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
