# Peer review of "Identification of Genomic Variants and Candidate Genes for Reproductive Traits and Growth Traits in Pishan Red Sheep Using Whole-Genome Resequencing"

_biology, 2025, doi:10.3390/biology14060636_

Round 1

Reviewer 1 Report

Comments and Suggestions for Authors

Dear Authors,

In the section on data collection in the material method, it was only mentioned that the sheep were divided into two groups according to their birth rate, but no information was given as to which body measurements were taken.

Line 63: It might be more appropriate to write “between birth and 1 months of age” instead of  “between 0 and 1 months of age”

Line 112: It may be more appropriate to write “aged 3-4 year” instead of “of the 3~4 age”.

Line 311: It may be more appropriate to write “birth” instead of “0”.

Figure 5C: I think “LR” is misspelled, it should be “LS”. No unit specified for LR (such as cm)

Author Response

Dear editors and reviewers:

Thank you very much for your valuable suggestions on this manuscript. Your suggestions and opinions have a good impact on our further revision and improvement of the manuscript. We carefully revised the manuscript according to the reviewer and provided the manuscript. All changes to the manuscript are point-by-point responses to the reviewers' comments in highlighted and nonhighlighted copies. The major revisions and responses to the reviewers' comments are as follows:

Reviewers' comments:

Reviewer 1

Comments to the Author

  1. In the section on data collection in the material method, it was only mentioned that the sheep were divided into two groups according to their birth rate, but no information was given as to which body measurements were taken.

 Response: Thank you very much for your professional comments and careful work. The phenotypic data of growth traits (i.e., body weight, body height, body length, chest circumference, cannon circumference, tail length, tail width and tail thickness) was carried out in accordance with the provisions of NY/T 1236. (L124-127, Marked red, Revised manuscript)

  1. Line 63: It might be more appropriate to write “between birth and 1 months of age” instead of  “between 0 and 1 months of age”

Response: Thank you very much for your Thank you very much for your kind comments and careful work. Our English is not very good. Unprofessional statements was revised. (L69-70, Marked red, Revised manuscript)

  1. Line 112: It may be more appropriate to write “aged 3-4 year” instead of “of the 3~4 age”.

Thank you very much for your Thank you very much for your kind comments and careful work. Unprofessional statements was revised.  (L119, Marked red, Revised manuscript)

  1. Line 311: It may be more appropriate to write “birth” instead of “0”.

Thank you very much for your Thank you very much for your kind comments and careful work. The “ female lambs between 0 and 30 days of age” was revised to “female lambs between birth and 30 days of age”. (L342, Marked red, Revised manuscript)

  1. Figure 5C: I think “LR” is misspelled, it should be “LS”. No unit specified for LR (such as cm)

Response: Thank you very much for your careful work. The litter size was indeed misspelled, the abbreviation was LR. The Fig.5C was revised.

Reviewer 2 Report

Comments and Suggestions for Authors

Population Structure Control: The PCA and ADMIXTURE results show limited differentiation between the HF and LF groups, raising concerns about population stratification. Although MLM models were used, the manuscript would benefit from additional justification or the application of stricter models to ensure robustness in GWAS analysis.

Selective Sweep Detection: The study uses top 5% FST and π ratio thresholds for selective sweep analysis. Incorporating complementary methods (e.g., XP-EHH, iHS) could strengthen the identification of true selection signatures.

Functional Validation: While candidate genes are identified through bioinformatic analysis, no experimental validation is provided. Even limited expression analysis would substantially enhance the biological significance of the findings.

Figure Legends: Some figure legends (e.g., Figure 5, Figure 6) are not sufficiently detailed, making it difficult for readers to interpret the subpanels independently. Please expand all figure legends to fully describe the content and findings of each panel.

Image Quality: Several figures appear to have relatively low resolution, making key details and text labels difficult to read. Please improve the image clarity and ensure that all figures meet high-resolution publication standards.

Candidate Gene Annotation: Tables listing GWAS results (e.g., Table 1) would be more informative if brief functional descriptions of the associated candidate genes were included alongside the gene names.

Other Comments: A careful proofreading for minor inconsistencies (e.g., "HF" and "LF" sometimes being introduced without redefinition) is recommended.

The discussion could benefit from a brief acknowledgment of the limitations related to environmental factors or other non-genetic influences on the traits.

Comments on the Quality of English Language

Some grammatical inconsistencies and occasional awkward phrasing are present. A careful proofreading is recommended to ensure smooth and concise expression.

Minor inconsistencies exist in the use of abbreviations (e.g., definitions of "HF" and "LF" should be consistently introduced when first mentioned in different sections).

The figure legends and table titles could be expanded for greater detail and independent readability.

Author Response

Reviewer:2

Comments and Suggestions for Authors

  1. Population Structure Control: The PCA and ADMIXTURE results show limited differentiation between the HF and LF groups, raising concerns about population stratification. Although MLM models were used, the manuscript would benefit from additional justification or the application of stricter models to ensure robustness in GWAS analysis.

Response: Thank you very much for your professional comments. For GWAS analysis, GLM model, MLM model and Farm CPU model in GAPIT package in R were used. The results of the three models are consistent, so in the manuscript, we only show the  MLM result. Now the corresponding content was supplemented and modified in the revised manuscript. (L176, L186-198, L264-266, Marked red, Revised manuscript)

  1. Selective Sweep Detection: The study uses top 5% FST and π ratio thresholds for selective sweep analysis. Incorporating complementary methods (e.g., XP-EHH, iHS) could strengthen the identification of true selection signatures.

Response: Thank you very much for your professional comments and careful work. We sincerely appreciate the reviewer’s valuable suggestions. In this study, we identified candidate selection regions using thresholds for FST and π ratio, two widely adopted metrics for detecting population differentiation and selection signals. The complementary use of low/high π ratio and high FST helps balance sensitivity and specificity in identifying selection signatures. We fully agree with the reviewer that haplotype-based methods such as XP-EHH and iHS could provide additional insights into recent or localized selection events, thereby strengthening the reliability of the findings. Due to the current data limitations the sample size. At present, the sample size is only 219 sheep, and other members of the project team have collected 200 Pishan red sheep for genome-wide analysis. So we opted not to include additional approaches at this stage. However, we plan to integrate cross-validation with multiple algorithms in follow-up work to further refine the detection and interpretation of selection signatures. We thank the reviewer again for their constructive feedback. 

  1. Functional Validation: While candidate genes are identified through bioinformatic analysis, no experimental validation is provided. Even limited expression analysis would substantially enhance the biological significance of the findings.

Response: Thank you very much for your professional comments. We used MassArray technology to verify the selected differential SNPs. Since this part of data is currently used to declare Chinese invention patents, there is no content on verification in the manuscript. In the future, we will add the verification of SNP in this part and verify the regulatory mechanism that functional genes may participate in at the cellular level.

  1. Figure Legends: Some figure legends (e.g., Figure 5, Figure 6) are not sufficiently detailed, making it difficult for readers to interpret the subpanels independently. Please expand all figure legends to fully describe the content and findings of each panel.

Response: Thank you very much for your valuable comments. The figure legends were revised. (Marked red, Revised manuscript)

  1. Image Quality: Several figures appear to have relatively low resolution, making key details and text labels difficult to read. Please improve the image clarity and ensure that all figures meet high-resolution publication standards.

Response: Thank you very much for your kind comments. The origin image were uploaded.

  1. Candidate Gene Annotation: Tables listing GWAS results (e.g., Table 1) would be more informative if brief functional descriptions of the associated candidate genes were included alongside the gene names.

Response: Thank you very much for your kind comments. The function of the genes were added.  (Table 1, Marked red, Revised manuscript)

  1. Other Comments: A careful proofreading for minor inconsistencies (e.g., "HF" and "LF" sometimes being introduced without redefinition) is recommended.

Response: Thank you very much for your kind comments and careful work. The definitions of HF and LF groupings were explained in the material method, so they were not explained again in subsequent results and discussions. While we were also revised. (L222, L223, Marked red, Revised manuscript)

  1. The discussion could benefit from a brief acknowledgment of the limitations related to environmental factors or other non-genetic influences on the traits.

Response: Thank you very much for your kind comments. The reason was added. (L326-327, Marked red, Revised manuscript)

  1. Comments on the Quality of English Language

Some grammatical inconsistencies and occasional awkward phrasing are present. A careful proofreading is recommended to ensure smooth and concise expression.

Response: Thank you very much for your kind comments. Our manuscript was revised by Charlesworth Author services. If there are still language problems in our manuscript, we are willing to modify it again in the author service of MDPI.

  1. Minor inconsistencies exist in the use of abbreviations (e.g., definitions of "HF" and "LF" should be consistently introduced when first mentioned in different sections).

Response: Thank you very much for your kind comments. The figure legend and table detail were revised. (L230, Marked red, Revised manuscript)

  1. The figure legends and table titles could be expanded for greater detail and independent readability.

Response: Thank you very much for your kind comments. The figure legend and table detail were revised. 

Reviewer 3 Report

Comments and Suggestions for Authors

The paper deals with the genomics of sheep. It is well made and brings new knowledge. Some revisions are needed.

Simple Summary

…single genetic variations… are SNPs. Use common terminology.

Abbreviations of genes in italic, check throughout the text.

Delete the last sentence rr. 20-21, the same Abstract rr. 37-38.

Introduction

Add into the 1st paragraph body mass of rams and ewes, wool production per year, litter size or percentage of twins, fertility per ewe and year, number of Pishan Red Sheep in China, in which provinces, etc. Some of the information is in Discussion, complete it in this section, or move it into Introduction.

Results

There was not possible to download the Supplementary data. Why?

Fig. 2, give it in better magnification, it is not well readable.

The same Figs. 3, 7.

Fig. 4 is slightly better.

Fig. 5., add legend, explain the abbreviations. Test the differences between LF and HF groups and add the significancy into the figure.

Fig. 6, explain the abbreviations.

Formal errors

  1. …32, pathwasy.˽A…
  2. 62 Lan et al. (point).

Author Response

Reviewer:3

Comments and Suggestions for Authors

  1. The paper deals with the genomics of sheep. It is well made and brings new knowledge. Some revisions are needed.

Simple Summary

…single genetic variations… are SNPs. Use common terminology.

Response: Thank you very much for your kind comments. The ‘single genetic variations’ was revised to ‘SNPs’. (L13, Marked red, Revised manuscript)

  1. Abbreviations of genes in italic, check throughout the text.

Response: Thank you very much for your kind comments. All the gene name were italic.

  1. Delete the last sentence rr. 20-21, the same Abstract rr. 37-38.

Response: Thank you very much for your kind comments. The sentences were deleted.

Introduction

  1. Add into the 1st paragraph body mass of rams and ewes, wool production per year, litter size or percentage of twins, fertility per ewe and year, number of Pishan Red Sheep in China, in which provinces, etc. Some of the information is in Discussion, complete it in this section, or move it into Introduction.

Response: Thank you very much for your professional comments. The appearance characteristics, inventory and other information of Pishan red sheep was supplemented. (L45-55, Marked red, Revised manuscript)

Results

  1. There was not possible to download the Supplementary data. Why?

Response: Thank you very much for your kind comments. All the Supplementary data were uploaded again.

  1. 2, give it in better magnification, it is not well readable.

Response: Thank you very much for your kind comments. Fig.2 was revised. And all the origin images were uploaded.

  1. The same Figs. 3, 7.

Response: Thank you very much for your kind comments. Fig.3 and Fig.7 were revised.

  1. 4 is slightly better.

Response: Thank you very much for your positive comments. Fig.4 was also revised.

  1. 5., add legend, explain the abbreviations. Test the differences between LF and HF groups and add the significancy into the figure.

Response: Thank you very much for your kind comments. Fig.5 was revised.

  1. 6, explain the abbreviations.

Response: Thank you very much for your kind comments. The abbreviations was explained in Fig.5

  1. Formal errors

…32, pathwasy.˽A…

62 Lan et al. (point).

Response: Thank you very much for your kind comments and careful work. The errors were revised. (L31, L68, Marked red, Revised manuscript)
